# Psychometric properties and longitudinal measurement invariance of the drug craving scale: Modification of the Polish version of the Penn Alcohol Craving Scale (PACS)

**Sylwia Opozda-Suder** [ORCID]⊕*, **Kinga Karteczka-Świętek**⊕, **Małgorzata Piasecka**

Institute of Education, Faculty of Philosophy, Jagiellonian University in Kraków, Kraków, Poland

⊕ These authors contributed equally to this work.
* sylwia.opozda@uj.edu.pl

## Abstract

### Background

The Penn Alcohol Craving Scale (PACS) is an instrument with good psychometric properties that is widely used to assess alcohol craving. Based on the assumption that the experience of craving is independent of substance type, the Polish version of the PACS was modified to measure drug craving, thus creating the Penn Drug Craving Scale (PDCS). The analyses presented in the paper aim to verify the hypothesis that the PDCS has a unidimensional structure, is highly reliable and features longitudinal measurement invariance.

### Methods

The research was conducted in 14 inpatient and 13 outpatient randomly selected facilities that provide psychosocial therapy to people with substance use disorder (SUD) in Poland, during June 2018 –July 2019. The data used for the analyses came from 282 patients diagnosed on the basis of ICD-10 criteria (F11.2-F19.2). The paper presents analyses with the application of: [1] confirmatory factor analysis (CFA) conducted on the basis of a polychoric correlation matrix and the WLSMV estimator; [2] a reliability estimate using Cronbach's alpha and coefficient omega; [3] verification of longitudinal measurement invariance between the beginning and end of therapy; [4] evaluation of criterion validity; [5] normalisation of the raw scores.

### Results

The CFA results confirmed a unidimensional PDCS structure (RMSEA = 0.047, 95% CI: 0.000–0.103; CFI = 0.999; TLI = 0.999) and a high reliability of the scale ($\omega$ = 0.93). Moreover, a strict longitudinal measurement invariance of the instrument was confirmed.

### Conclusions

Accurate assessment of craving is possible only with valid and reliable instruments. Therefore, the psychometric properties of the PDCS were verified based on the latest statistical

**Funding:** The research was conducted as part of a project selected in a competition procedure of the National Bureau for Drug Prevention (www.kbpn. gov.pl). Resources from the Fund for Solving Gambling Problems, which are available to the Minister of Health, constituted the source of its co-financing (140/W/H4/2018; 221/BO/H4/2019). The funds were awarded for the project to the Jagiellonian University; they were not assigned to a specific person. Author SOS received funding for a publication in Open Access in a competition procedure. The publication was funded by the Priority Research Area Society of the Future under the program "Excellence Initiative – Research University" at the Jagiellonian University in Krakow (www.futuresoc.id.uj.edu.pl) (166.0641.363.2019). The funders had no role in the study design, data collection and analysis, decision to publish, or preparation of the manuscript.

**Competing interests:** The authors have declared that no competing interests exist.

approaches. The scale is a valid and highly reliable tool featuring longitudinal measurement invariance and can be usefully used for research and clinical purposes. Thus, the Polish version of the PACS has been modified and successfully applied to the population of people with SUD.

## Introduction

Craving is a central notion in numerous modern models of addiction in relation to all psychoactive substances [1, 2]. Based on multiple research studies [3–6], craving was re-included in the DSM-5 [7] as one of the diagnostic criteria of substance use disorders.

According to a review of the craving literature, craving is defined as a subjective experience, an individual state that motivates the use of a substance, it can be simply described as a *desire to use* a substance [1, 2, 8–10]. It is a conscious, strong desire or a sensation of a compulsion to take a specific substance [2, 11]. As a multi-dimensional construct, its scope exceeds the category of an urge. It encompasses the prediction of positive effects of use, the urge to alleviate abstinence symptoms by use, and targeted activity directed at obtaining and using a psychoactive substance [2, 12–15]. For a person with substance use disorder (SUD) craving is an aversive experience which disturbs their functioning [2]; it is particularly burdensome due to its dynamic and fluctuating nature [2, 16]. The theory distinguishes phasic and tonic craving. Phasic craving means relatively short desire escalations in response to environmental factors or emotionally loaded stimuli, which remind a person with SUD of the episodes of usage or signal an approaching opportunity to do so. On the other hand, tonic craving is a retrospective, subjective and slowly changing experience of a desire which lasts for a specific period of time in a situation where it has not been triggered [2, 16, 17].

Reducing craving and developing an ability to cope with it constitute the major elements of most therapies [18–24]. As a result, verification of the clinical utility of craving—treated as an object of therapeutic intervention—is the subject of numerous studies [3–6, 25, 26]. Hence there is a need for a rigorous evaluation of the psychometric and predictive properties of self-report instruments for measuring substance craving. Accordingly, the research team at the Institute of Education at the Jagiellonian University, in their scientific project on the direct and delayed effects of therapy for people with SUD, considered analyses relating to substance craving [27]. A review of the measurement tools available in Poland conducted at the time led to the conclusion that the only scale with verified psychometric properties is an adaptation of the Penn Alcohol Craving Scale (PACS) by Chodkiewicz et al. [28]. Based on the assumption that the experience of craving is independent of substance type, the team modified the PACS for drug craving measurement. The accuracy of this approach was confirmed by other researchers' attempts to adapt the PACS to study craving in relation to opiates or methamphetamine addiction [29–31].

## Aims of the study

This paper aims to present the psychometric properties of the Penn Drug Craving Scale (PDCS), which is a modification of a Polish adaptation of the PACS [28]. Accordingly, the manuscript presents the results of verification of the hypothesis that the PACS, when adapted for drug craving measurement, maintains its unidimensional structure as well as a high reliability level. Additionally, the assessment of the longitudinal measurement invariance of the PDCS enabled a verification of the hypothesis that regardless of the time when a measurement

is taken (therapy stage), the scale measures drug craving with the same precision; therefore it has the same level of measurement reliability over time.

## Methods

### Research procedures

The PDCS research was conducted as part of a broader research project. This project was approved by the Ethics and Research Committee operating at the Jagiellonian University.

The research was carried out in 14 inpatient and 13 outpatient randomly selected facilities for the psychosocial treatment of people with SUD in Poland. In most facilities participating in the research, the treatment was based on cognitive-behavioural psychotherapy, modified mostly by combining it with methods such as motivational interviewing, therapeutic community or solution-focused brief therapy. Average therapy duration was 6 months, with a range from 2 to 12 months.

The research was conducted from June 2018 until July 2019. Once records with missing data regarding responses to the PDCS questions were removed, the analyses were conducted on 282 cases. The data collection process was developed during two studies. In Study 1, data were collected from 111 patients at different stages of the therapy. Study 2 was longitudinal and consisted of measurements at two time points (T1 and T2). T1 was conducted among patients at the beginning of the therapy (where the beginning of therapy means that the patients had been under treatment in a particular facility for no longer than two weeks). 171 patients were surveyed at T1. At T2, data were collected from 70 out of these patients who had completed their therapy (the rest failed to complete the therapy).

All patients were informed in writing about the purpose and principles of the study, and, on this basis, they gave written consent to participate. The study was conducted with a traditional pen-and-paper method. Surveys were carried out by addiction therapists trained for this purpose.

### Participants

The respondents were adults (over the age of 18) who were Polish nationals with a substance use disorder diagnosis in accordance with the ICD-10 criteria (F11.2-F19.2). Almost 87% of them were male (this proportion corresponds to the specifics of the Polish population of people with SUD). The respondents' average age was 27.40 years, within the range from 18 to 48. Detailed characteristics of the sample are included in Table 1.

### Instruments

*The Penn Drug Craving Scale (PDCS)* used in the research is the Polish version of the Penn Alcohol Craving Scale (PACS) modified for drug craving measurement.

The PACS is one of the most common instruments for assessing the intensity of tonic alcohol craving [9, 17, 32, 33]. Research studies that have employed the PACS indicate that this instrument demonstrates a high prognostic utility for a risk of relapse and this exceeds the effect achieved with the application of other instruments [5, 34].

The attempt to adapt the content of the PACS to measure drug craving was based on the principle that wherever there was terminology about alcohol it was replaced with words connected with drugs. Therefore, the questions in the PDCS sound almost identical to those in the PACS [35]. The changes between the tools are presented in detail in the S1 Appendix. Additionally, since the current research was conducted on a Polish-language sample, the PDCS in Polish is included in the S2 Appendix.

**Table 1. Descriptive characteristics of sample.**

| Characteristic | n | % |
|---|---|---|
| SOCIO-DEMOGRAPHIC | | |
| Male | 244 | 86.52 |
| Mean age in years (SD) | 27.40 | (6.57) |
| Civil status | | |
| Single | 178 | 63.12 |
| Married | 31 | 10.99 |
| Divorced or separated | 22 | 7.80 |
| With a partner | 51 | 18.09 |
| Completed high school | 193 | 68.44 |
| EMPLOYMENT STATUS | | |
| Unemployed | 122 | 43.26 |
| Median monthly income[1] | 2,200 | |
| ADDICTION CHARACTERISTICS | | |
| Mean age of first use (SD) | 16.20 | (3.36) |
| Other addictions[2] | 162 | 57.45 |
| The most frequently used drug[2] | | |
| Cannabis | 128 | 45.39 |
| New psychoactive substances (NPS) | 46 | 16.31 |
| Amphetamines and other stimulants | 74 | 26.24 |
| Type of therapy | | |
| Inpatient (residential) | 195 | 69.15 |
| Mean duration of therapy in months (SD)[3] | 4.79 | (1.75) |

Note.

[1] n = 164, number of people with income; national average monthly income = 3,775 PLN;

[2] self-reported;

[3] assumed at the beginning of the therapy

Both the PACS and the PDCS consist of 5-items of a self-report type and provide a comprehensive result. The answer options consist of various statements that are rated on a 7-point scale. Individual questions refer to information on frequency of craving, intensity, duration of thinking about taking a drink/drug, ability to resist drinking/taking a drug and the general urge throughout the past week. Consequently, the diagnostics spectrum of the tools is more comprehensive than that of most other instruments that are used to assess craving intensity alone [2]. The value of Cronbach's alpha in the original English version [35] and the Polish version of the PACS [28] equals 0.91 and 0.89, respectively.

In the research, Polish adaptations of the following widely-known, reliable instruments were applied: the Buss-Perry Aggression Questionnaire (BPAQ) [36]; the Impulsiveness and Venturesomeness Questionnaire (IVE) [37]; the Achievement Motivation Inventory (LMI) [38]; the Generalized Self-Efficacy Scale (GSES) [39]; and the Loneliness Scale (R-UCLA) [40]. Additionally, the original Drug Desire Scale (SPN) and Self-Aggression Scale (SAA) [27] were used. The criterion validity of the PDCS was evaluated on the basis of the results from all the mentioned questionnaires.

## Data analysis

The assessment of the psychometric properties of the PDCS was primarily aimed at confirming its unidimensional structure, which characterizes the PACS [28, 35]. Therefore, a confirmatory

factor analysis (CFA) was applied. The CFA was conducted based on a polychoric correlation matrix. The model was estimated with the WLSMV *(weighted least squares means and variance adjusted)* estimator [41]. The CFA model fit was assessed on the basis of three indices: RMSEA *(root mean square error of approximation)*, CFI *(comparative fit index)* and TLI *(Tucker-Lewis index)*. It was assumed that a model that fitted well to the data would be one where the value of RMSEA < 0.08, while CFI and TLI > 0.90 [42].

The reliability of the PDCS was determined based on Cronbach's alpha and the coefficient omega (ω) calculated using the parameters estimated for the model. For both measures, it was assumed that a value > 0.80 is an indicator of high reliability of the tool [43, 44].

In order to verify the reliability over time of the PDCS, the longitudinal measurement invariance (LMI) was examined between the beginning (T1) and the end of therapy (T2). Configural, metric, scalar and strict invariances were tested respectively. In the case of configural invariance, it was assumed that the fundamental model structure in T1 and T2 is the same, and all parameters of the model may differ between the two time points. In the case of metric invariance, it was assumed that the factor loadings do not differ between T1 and T2, whereas for scalar invariance, the thresholds do not differ either. In the case of strict invariance, another constraint was imposed on the scalar model, involving the equalisation of residual variances for individual items. To verify whether the restrictions imposed onto individual models worsen their fit in the case of the chi-square test, the DIFFTEST procedure from the Mplus package [45] was applied. On the other hand, the difference in values (Δ) was calculated for the RMSEA and the CFI measures. Measurement invariance was confirmed when ΔRMSEA ≤ 0.007, and ΔCFI ≥ -0.002 [46].

A criterion validity analysis was also conducted by determining the value of the r-Pearson correlation coefficient between the PDCS result and the results from other tools, constituting the comparative criteria.

A normalisation of the PDCS results—due to the skewed character of their distribution—was prepared using a tercile scale. A tercile scale does not reflect the shape of the raw score distribution; the distribution of its values is always uniform. This means there is the same probability of the occurrence of all values of a variable.

For all analyses involving a probability value, 0.05 was assumed as the threshold for statistical significance. In the presentation of the results of analyses in which a p-value was needed, it was reported each time.

The modelling was performed with Mplus 8.3 [45]. The reliability and criterion validity analysis were conducted using RStudio 1.2.5. with the application of the lavaan package [47]. Furthermore, Jasp 0.12.2 statistical software [48] was used for other analyses.

## Results

### Confirmatory factor analysis

The unidimensional model structure of the PDCS tested under the CFA has very good fit parameters, as is indicated by the following statistics: $\chi^2$ (5) = 8.095, p = 0.151, RMSEA = 0.047, 95% CI: 0.000–0.103, CFI = 0.999 and TLI = 0.999. Thus, a single-factor structure of the scale (one latent variable measured with 5 items) was confirmed.

Table 2 presents the values of standardised factor loadings for the estimated model as well as the PDCS items content. The main factor (latent variable) significantly loads on all scale items (p < 0.001). Furthermore, the loadings are very high, the average result is 0.86. Only in the case of item 4 does the achieved value slightly deviate from the remaining ones (λ = 0.70).

**Table 2. Standardised factor loadings: Confirmatory factor analysis.**

|  | Test question | Factor loading ($\lambda$) |
|---|---|---|
| item 1 | How often have you thought about taking drugs or about how good taking drugs would make you feel during this period? | 0.876*** |
| item 2 | At its most severe point, how strong was your craving during this period? | 0.868*** |
| item 3 | How much time have you spent thinking about taking drugs or about how good taking drugs would make you feel during this period? | 0.905*** |
| item 4 | How difficult would it have been to resist taking drugs during this period of time if you had known the drugs were in your house? | 0.701*** |
| item 5 | Keeping in mind your responses to the previous questions, please rate your overall average drug craving for the stated period of time. | 0.930*** |

*** indicates $p < 0.001$

## Reliability

The reliability of the PDCS was determined using coefficient omega ($\omega$). This coefficient indicates to what extent the general score for the tool may be interpreted as an indicator of drug craving intensity. Based on the achieved value, $\omega = 0.93$, one may conclude that the PDCS is a highly reliable tool, reflecting the variance of the latent variable. This conclusion is also confirmed by the value of a classic measure of internal consistency: Cronbach's alpha ($\alpha = 0.93$). Due to the fact that $\omega > 0.80$, the general score can be considered as the result of one factor [49]. This confirms the hypothesis that the PDCS is a strictly unidimensional scale. Furthermore, the main factor explains as much as 93% of the variance of the general score on the scale.

## Longitudinal measurement invariance

The analysis of the LMI followed the assessment of differences in the respondents' response distribution for each questionnaire item between T1 and T2. A graphic method of data presentation, a violin plot, was applied for this purpose (Fig 1).

Fig 1 shows that for responses at T1 the achieved results are more broadly spread throughout the whole scale. On the other hand, at T2 the data for each item show a narrow spread concentrated around low values. Furthermore (at T2), for items 1, 3 and 5, it was highly unlikely that the respondent would mark an answer higher than 2 –corresponding to mild craving. At the same time, it is notable that for item 5 –at T1 as well as at T2 –the answers provided are within the range from 0 to 5, which means that none of the respondents indicated the maximum craving intensity. It is significant that the box plots for the results from all items at T2 do not contain a visible bottom whisker, which means that over 25% respondents indicated an answer declaring a lack of craving (0 on the answer scale). Additionally, the median is 1; therefore at least half of the respondents defined their craving as very mild at most.

Table 3 contains descriptive statistics for T1 and T2 scores alongside a result of the Wilcoxon signed-rank test (Z = -5.661, $p < 0.001$). The outcomes indicate significantly lower results at T2 than at T1. This confirms the need to resolve the issue of whether the existing differences stem from actual changes in the drug craving intensity or whether they are the effect of a lack of reliability over time of the PDCS. An answer to this question is provided on the basis of the LMI results, which are presented in Table 4.

The verification of the LMI started with the determination of the configural invariance. The configural model was well fitted to the data (RMSEA = 0.065, CFI = 0.998). Therefore, the

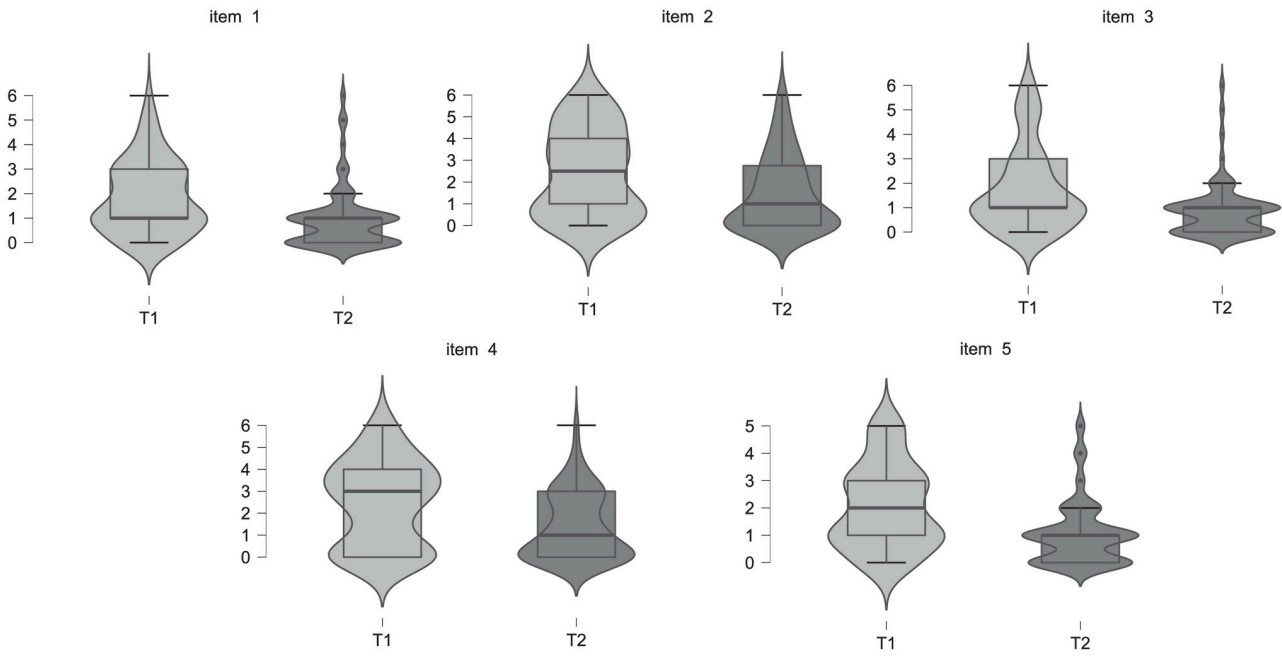

**Fig 1. Violin plot for each item of the PDCS: Longitudinal perspective (across T1 and T2).**

assessment of the metric model commenced, and it proved to be equally well fitted to the data as the configural model. The achieved values, ΔRMSEA = 0.005 and ΔCFI > -0.001, meet the assumptions for the adopted limit values, and they confirm the hypothesis on the metric measurement invariance of the PDCS.

Testing the scalar invariance did not cause any deterioration of the fit of the model either. The Δ values for the fit indices here also meet the criteria (ΔRMSEA = -0.005, ΔCFI = -0.001). Hence, the PDCS displays scalar invariance. Therefore, it is reasonable to compare mean latent variable (drug craving) values obtained during consecutive measurements.

In the final stage of the analysis of the LMI, strict invariance was verified. The estimated model does not provide a basis for rejecting the hypothesis on the strict invariance of the scale. This is indicated in the results of the DIFFTEST ($\Delta\chi^2$ (28) = 40.977, p = 0.333), as well as

**Table 3. Descriptive statistics for the PDCS (across T1 and T2).**

| PDCS (n = 70) | T1 | T2 |
|---|---|---|
| Mean [95% CI] | 10.56 [8.72–12.39] | 5.80 [4.37–7.23] |
| Std. Deviation | 7.70 | 6.01 |
| Median | 9.00 | 4.50 |
| Skewness[1] | 0.38 | 1.25 |
| Kurtosis[2] | -0.93 | -1.58 |
| Shapiro-Wilk | 0.95 | 0.86 |
| p-value | 0.004 | < 0.001 |
| Wilcoxon | Z = -5.661 | |
| p-value | < 0.001 | |

[1] S.E. of Skewness = 0.29;

[2] S.E. of Kurtosis = 0.57;

**Table 4. Longitudinal invariance: Models' fit (across T1 and T2).**

| MODEL | Chi-Square Test[a] | | | | RMSEA | ΔRMSEA | CFI | ΔCFI |
|---|---|---|---|---|---|---|---|---|
| | $\chi^2$ | df | $\Delta\chi^2$ | Δdf | | | | |
| Configural[1] | 40.081[#] | 31 | – | – | 0.065 | – | 0.998 | – |
| Metric[2] | 43.854[#] | 35 | 3.315[#] | 4 | 0.060 | 0.005 | 0.998 | -0.000 |
| Scalar[3] | 70.234[#] | 58 | 28.260[#] | 23 | 0.055 | -0.005 | 0.997 | -0.001 |
| Strict[4] | 83.375 | 63 | 40.977[#] | 28 | 0.068 | 0.013 | 0.995 | -0.002 |

[#] indicates p > 0.05;

Note.

[a] $\Delta\chi^2$ and Δdf calculated with the DIFFTEST procedure from the Mplus package;

[1] configural invariance—all parameters of the model may differ between T1 and T2;

[2] metric invariance—factor loadings are equal between T1 and T2;

[3] scalar invariance—factor loadings and thresholds are equal between T1 and T2;

[4] strict invariance—factor loadings, thresholds and residual variances are equal between T1 and T2.

ΔCFI = -0.002. Only in the case of ΔRMSEA is the achieved value, 0.013, higher than the assumed threshold 0.007. However, it is a highly restrictive criterion. An additional argument for the strict measurement invariance of the scale is the convergence of the value of the omega (ω) coefficient calculated separately for T1 and T2 ($\omega_{T1}$ = 0.955, $\omega_{T2}$ = 0.961). This is important because the essence of strict invariance is close measurement consistency at both time points. The data are completed with very similar values of factor loadings at T1 and T2 for individual items in the strict invariance model. Table 5 presents loadings estimates for all the invariance models tested.

The presented analyses demonstrate that the PDCS is an instrument that displays strict LMI. Achieving the highest invariance level means that regardless of the time of collection of responses, the scale measures a latent variable with the same error level (the same precision) and hence displays the same level of measurement time reliability. In consequence, when using the PDCS it is possible to compare the results across times, considering the observed differences to be the effect of an actual change of drug craving intensity.

## Criterion validity

The assessment of criterion validity is always based on an analysis of relations. In publications addressing the issue of a correlation between craving and predicting a relapse, craving is

**Table 5. Longitudinal invariance: Factor loadings (across T1 and T2).**

| MODEL | Factor loadings (T1 / T2) | | | | |
|---|---|---|---|---|---|
| | item 1 | item 2 | item 3 | item 4 | item 5 |
| Configural[1] | 0.923 / 0.977 | 0.932 / 0.853 | 0.927 / 0.954 | 0.750 / 0.781 | 0.950 / 0.977 |
| Metric[2] | 0.924 / 0.977 | 0.932 / 0.853 | 0.926 / 0.954 | 0.748 / 0.782 | 0.951 / 0.977 |
| Scalar[3] | 0.930 / 0.971 | 0.931 / 0.854 | 0.925 / 0.955 | 0.742 / 0.792 | 0.950 / 0.979 |
| Strict[4] | 0.953 / 0.957 | 0.892 / 0.900 | 0.939 / 0.944 | 0.761 / 0.775 | 0.968 / 0.970 |

Note.

[1] configural invariance—all parameters of the model may differ between T1 and T2;

[2] metric invariance—factor loadings are equal between T1 and T2;

[3] scalar invariance—factor loadings and thresholds are equal between T1 and T2;

[4] strict invariance—factor loadings, thresholds and residual variances are equal between T1 and T2.

**Table 6. Correlation coefficient of the PDCS and the results of other research instruments[1].**

|  | SPN | BPAQ | BPAQ-A | BPAQ-H | SAA | IVE | R-UCLA | AMI | GSES |
|---|---|---|---|---|---|---|---|---|---|
| PDCS | 0.41*** | 0.29** | 0.25* | 0.28** | 0.27** | 0.27** | 0.24* | -0.29** | -0.19* |
|  | [.25,.55] | [.09,.47] | [.05,.42] | [.09,.45] | [.09,.43] | [.10,.43] | [.04,.41] | [-.45, -.12] | [-.36, -.01] |

[1] n = 111;

* indicates p < 0.05.

** indicates p < 0.01.

*** indicates p < 0.001.

Note. Values in square brackets indicate the 95% confidence interval for each correlation. The confidence interval is a plausible range of population correlations that could have caused the sample correlation.

SPN—Drug Desire Scale; BPAQ—Buss-Perry Aggression Questionnaire; BPAQ-A—Anger Scale; BPAQ-H—Hostility Scale; SAA—Self-Aggression Scale; IVE—Impulsiveness and Venturesomeness Questionnaire; R-UCLA—Loneliness Scale; AMI—Achievement Motivation Inventory; GSES—Generalized Self-Efficacy Scale.

shown as a co-determining factor, alongside other intra- and interpersonal variables such as self-efficacy [50–52], motivation [53–56], negative affect (aggression, self-aggression—self-injury, impulsiveness) [57–62] and social relations (sense of loneliness, social support) [63–66]. Most of the reported relations are incorporated in the cognitive-behavioural model of relapse [59]. All of the listed factors contribute to a relapse; therefore their mutual interactions are also assumed. Based on this assumption, these variables were considered comparative criteria. The criterion validity assessment involved the measurement of the relations of the PDCS with instruments testing criterion variables and other scale assessing craving.

The PDCS features satisfactory criterion validity. This is indicated by statistically significant correlations between the observed general score of the scale and the observed results from other tools used. Table 6 presents the obtained r-Pearson correlation coefficients.

The highest value was achieved for the result from the SPN (r = 0.41, p < 0.001), where craving was operationalised in a manner that is comparable to the PACS. Positive values of the correlation coefficient were obtained for criteria linked to a negative affect measurement. For a general result from the BPAQ, r = 0.29 (p = 0.005), and also for the SAA and the IVE, r = 0.27 (p = 0.003). A positive correlation, r = 0.24 (p = 0.016), was also observed for the R-UCLA, which is the only criterion referring to interpersonal determinants. Negative values of the correlation coefficient were obtained in the case of two cognitive constructs relating to achievement motivation (AMI) (r = -0.29, p = 0.001) and a self-efficacy (GSES) (r = -0.19, p = 0.042).

Notably, all correlations with the compared intra- and interpersonal features are statistically significant. The achieved coefficient values indicate weak correlations. Such results were expected and are justified in the relapse prevention model [59]. This model assumes that all the variables jointly determine the risk of relapse; however, at the same time it does not establish strong correlations between them.

## Percentile norms

In terms of the diagnostic utility of the PDCS, the final stage of the analyses was normalisation. Similarly to the PACS, the raw score on the PDCS is calculated by summing up the points from the respondents' answers to all five items. Points are assigned to answers to each question as follows: a value of 0 indicates an absence of drug craving and a value of 6 indicates strong intensification of drug craving. Then the raw score should be referred to a defined normalisation scale. Due to the confirmed skewness of the PDCS results distribution, tercile norms were prepared—similarly to the study by Chodkiewicz et al. [28]. The range of variability of the raw

score was categorized in the following manner: 0–4 –low craving intensity, 5–10 –average, over 10 –high.

The normalisation sample was diversified according to the therapy stage and therefore those norms may be applied regardless of the therapy stage a respondent is at. Furthermore, their application is justified only in the case of respondents from the population featuring characteristics similar to the normalisation sample (see Table 1).

The norms achieved are very close to those defined for the Polish version of the PACS [28]. When compared, the differentiating value of the first and the second tercile was shifted by +1, whereas the value of the third tercile corresponded completely. This allows us to conclude that the distribution of craving intensity in the population of people with substance use disorder is similar to that of the population of people with alcohol use disorder.

## Discussion

Progress on research into the clinical utility of craving can be achieved by studies confirming the validity and reliability of the measurement instruments applied [67]. Accordingly, the research undertaken should aim at verification of the psychometric properties of these instruments, using the latest analytical procedures, including those presented in this paper, namely analyses of the latent structure and longitudinal measurement invariance. This is particularly relevant for drug craving, as there is remarkably little research on the assessment of psychometric properties of the measurement scales compared to those used for alcohol craving.

In response to the needs stated above, this paper aimed to verify the psychometric properties and measurement utility of the Penn Drug Craving Scale (PDCS), which is a modification of the Polish adaptation of the Penn Alcohol Craving Scale (PACS).

Pointing to the strengths of the study, the PACS was successfully adapted for drug craving measurement using the latest analytical procedures. Based on the results achieved from the CFA, the unidimensional nature of the PDCS was confirmed. In comparison to the PACS, the reliability of the PDCS turned out to be slightly higher [28, 35]. The research conducted positively verified the hypothesis on the strict longitudinal measurement invariance of the PDCS. Achieving such a level of LMI—despite the varying length of therapy—is another strong point of this validation study. Thus, it is reasonable to compare the results obtained at particular stages of therapy and to recognize the observed differences in craving intensity as evidence of the therapy's effectiveness. Also the criterion validity of the PDCS can be considered satisfactory. The obtained results of correlation confirmed the correctness of the assumption that there is an indirect relation of craving with intra- and interpersonal factors which jointly determine the risk of a relapse.

At the same time, this study is not free of limitations. The criterion validity of the PDCS should be expanded by correlations with the results of other commonly known instruments for measuring drug craving. However, this first requires the creation of Polish adaptations of these tools. In accordance with the authors' best knowledge—the PACS is still the only tool that has been standardised and adapted to measure substance craving. Moreover, the research data collection process was not directed at sample differentiation in accordance with substance type used. Due to this, estimating multi-group measurement invariance was not feasible.

As for further research on use of the PDCS, three directions may be indicated. The application of the cognitive-behavioural model of relapse [59] to assess criterion validity indicates a need for research with the use of modelling of multivariable, complex relation systems between various factors and therapy effectiveness. Thus, the predictive power of each variable used, particularly drug craving, will be known. It also seems necessary to confirm the prognostic utility of the scale—in particular, attempting to determine the cut-off point of the PDCS score. This

would help predict with high probability that the expected therapy effects will not be met or relapse will occur. The final direction of further research with the application of the PDCS is the issue of verification of multi-group measurement invariance stemming from the type of psychoactive substance used.

## Conclusions

The analyses of the psychometric properties and longitudinal measurement invariance of the PDCS indicate that this tool may be successfully used to measure drug craving for research and clinical purposes. The PDCS could be particularly useful for continuous monitoring of therapy effectiveness, since a change in intensity of craving is a significant indicator (of its effectiveness). It enables a simple comparison of craving measurement results, without a need to involve advanced statistical methods, which is highly advantageous.

## Supporting information

**S1 Appendix. The changes between the PDCS and the PACS positions content.**
(PDF)

**S2 Appendix. Polish version of the Penn Drug Craving Scale (PDCS).**
(PDF)

**S1 Data. Confirmatory factor analysis.**
(DAT)

**S2 Data. Longitudinal measurement invariance.**
(DAT)

**S3 Data. Criterion validity.**
(DAT)

## Author Contributions

**Conceptualization:** Sylwia Opozda-Suder, Kinga Karteczka-Świętek, Małgorzata Piasecka.

**Formal analysis:** Sylwia Opozda-Suder, Kinga Karteczka-Świętek.

**Funding acquisition:** Sylwia Opozda-Suder, Kinga Karteczka-Świętek, Małgorzata Piasecka.

**Investigation:** Małgorzata Piasecka.

**Methodology:** Sylwia Opozda-Suder, Kinga Karteczka-Świętek.

**Project administration:** Małgorzata Piasecka.

**Visualization:** Sylwia Opozda-Suder, Kinga Karteczka-Świętek.

**Writing – original draft:** Sylwia Opozda-Suder, Kinga Karteczka-Świętek, Małgorzata Piasecka.

**Writing – review & editing:** Sylwia Opozda-Suder, Kinga Karteczka-Świętek.

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
