## [Decision Letter · Decision Letter 0]

16 Jun 2021

PONE-D-20-36075

Psychometric properties and longitudinal measurement invariance of the drug craving scale: modification of the Polish version of the Penn Alcohol Craving Scale (PACS)

PLOS ONE

Dear Dr. Opozda-Suder,

Thank you for submitting your manuscript to PLOS ONE. After careful consideration, we feel that it has merit but does not fully meet PLOS ONE’s publication criteria as it currently stands. Therefore, we invite you to submit a revised version of the manuscript that addresses the points raised during the review process.

We look forward to receiving your revised manuscript.

Kind regards,

Frantisek Sudzina

Academic Editor

PLOS ONE

Journal Requirements:

2. Please note that according to our submission guidelines (http://journals.plos.org/plosone/s/submission-guidelines), outmoded terms and potentially stigmatizing labels should be changed to more current, acceptable terminology. For example: "drug addict" should be changed to "person with alcohol use disorder,” or “person with substance use disorder”.

Reviewers' comments:

Reviewer's Responses to Questions

**Comments to the Author**

1. Is the manuscript technically sound, and do the data support the conclusions?

Reviewer #1: Partly

Reviewer #2: Yes

2. Has the statistical analysis been performed appropriately and rigorously? 

Reviewer #1: Yes

Reviewer #2: Yes

3. Have the authors made all data underlying the findings in their manuscript fully available?

Reviewer #1: Yes

Reviewer #2: Yes

4. Is the manuscript presented in an intelligible fashion and written in standard English?

Reviewer #1: Yes

Reviewer #2: Yes

5. Review Comments to the Author

Reviewer #1: The manuscript, “Psychometric properties and longitudinal measurement invariance of the drug craving scale: modification of the Polish version of the PACS” is an important contribution to the literature regarding the assessment of craving for drugs in diverse populations. Valid & reliable assessment is key for any symptom of substance use, particularly one as mercurial as craving. Extending the well-known PACS to a drug using population in Poland is an important step. The CFA and reliability analyses are appropriate and well interpreted. The manuscript would be improved through further characterization of the sample and explanation of the approach of assessing criterion validity. I appreciate that publishing in one’s non-native language presents unique challenges and the manuscript would also benefit from thorough editing for word choice and flow. My specific comments and questions are as follows:

1. Authors make an assumption that craving is a “universal experience” amongst those with a substance use disorder when it is fact widely recognized that it is not.

2. The paper seems to contract its own definition of craving (paragraph at top of pg. 3)

3. The introduction reads as somewhat meandering by discussing a host of topics related to craving. It would benefit from being streamlined to the topic at hand.

4. The description of when and how the participants were ascertained is confusing (pg. 5, paragraph beginning “The research was conducted from…”). What is meant by starting “therapy”? Was this psychosocial treatment only? Also, the manuscript says data was collected longitudinal, but the following sentence implies that different patients were assessed at different points in therapy.

5. In the “primary drug used” of Table 1, please provide what is being abbreviated by “NPS.” Does primary drug use correspond to the substance use disorder (SUD) diagnoses? Would it be possible to provide SUD diagnosis as well as drug of choice? Mean & SD of length of treatment would also be appropriate to include in this table.

6. Given the length of treatment was highly variable (between 2-12 months), would this impact the LMI results being compared for T1 & T2? Were the mid-therapy assessments not used in any analyses?

7. The measures used to assess criterion validity are not discussed in the methods section. Besides the SPN, which seems to be another measure of craving, how were the other measures picked? How do they demonstrate criterion validity? There is no rationale provided as to why those measures were chosen and I wouldn’t necessarily expect there to be strong correlation between PDCS and the majority of the measures presented in Table 7.

8. In the discussion, further commentary on the strengths/limitations of the study is warranted.

9. The findings of the study are somewhat overstated in the conclusion section. While the PDCS may be used clinically, this study does not “confirm its clinical utility” nor investigate whether this measure improves treatment planning or predicts relapse.

Reviewer #2: Feedback for the manuscript entitled “Psychometric properties and longitudinal measurement invariance of the drug craving scale: modification of the Polish version of the Penn Alcohol Craving Scale (PACS)”

The study was intended to explore and verify the hypothesis that the Polish Drug Craving Scale (PDCS) has a unidimensional structure and highly reliable and longitudinal measurement invariance features. In addition, the study also presented criterion validity and percentile norms of the scale. Overall, this study is very useful and can draw some attention to practical users who are interested in using the Polish Drug Craving Scale. However, the manuscript still needs some revisions and a proofread.

1. The Polish Drug Craving Scale (PDCS) should be mentioned in the title instead of PACS.

2. On page 4, some specific research questions can be developed to guide the reader to better understand the study. These questions can follow “Aims of the analysis”. In addition, “Aims of the analysis” can be renamed “Aims of This Study”.

3. In the methods, the authors missed the description of the PDCS (Table 3). How many items? How many points? Likert scale?

4. Line 148 on page 7, MPLUS needs a citation.

5. Lines 155-156 on page 7, please specifically indicate what tests need the p-values?

6. Lines 157-158 on page 8, please specifically indicate what analyses using MPLUS and LAVAAN, respectively.

7. Some statistical analyses in the results were not mentioned in the methods. The authors reported criterion validity and percentile norms of the scale. But I could not find any statements regarding these two results in the methods.

8. For criterion validity, I am not sure if the authors used latent scores or observed scores to correlate with criterion variables.

9. On page 8, since all the values are reported in the text, there is no need to present this table (Table 2). Please add 95%RMSEA in the text as well.

10. The longitudinal measurement invariance should be reported before the descriptive statistics at T1 and T2.

11. LMI results should be reported before the descriptive statistics. Because as the authors mentioned that “it is reasonable to compare latent variable means (drug craving) obtained during consecutive measurements” due to LMI of the scale.

12. Lines 239-240, the statistical values for chi-square with df and p-value can be reported here.

13. Some statistics reports violate the APA style (p < .05 or p < .01 or p < .001)

6. PLOS authors have the option to publish the peer review history of their article (what does this mean?). If published, this will include your full peer review and any attached files.

Reviewer #1: No

Reviewer #2: No

---

## [Author Response · Author response to Decision Letter 0]

21 Jul 2021

We appreciate the thorough review and very constructive suggestions. After careful consideration, we have responded to each comment and heeded the reviewers’ feedback. The descriptions of the changes made are included below.

The line numbering given in the answers refers to the version of the text subjected to review (first submission).

ACADEMIC EDITOR:

Answer: We have checked and corrected the manuscript according to PLOS ONE's style requirements. 

2. Please note that according to our submission guidelines outmoded terms and potentially stigmatizing labels should be changed to more current, acceptable terminology. For example: "drug addict" should be changed to "person with alcohol use disorder,” or “person with substance use disorder”.

Answer: Thank you for the comment. We have changed problematic terms for acceptable terminology.

REVIEWER #1:

According to general comment that “the manuscript would also benefit from thorough editing for word choice and flow” our paper has been checked and corrected by a highly experienced native proofreader.

1. Authors make an assumption that craving is a “universal experience” amongst those with a substance use disorder when it is fact widely recognized that it is not.

Answer: Thank you for drawing attention to this issue. It seems to us that our choice of the word “universal” was misleading. We meant to indicate that craving is an experience which occurs independently of substance type. Therefore we have changed lines 16-17 to the following:

“Based on the assumption that the experience of craving is independent of substance type, the Polish version of the PACS was modified to measure drug craving, thus creating the Penn Drug Craving Scale (PDCS)”.

A similar correction applies to the sentence in lines 73-74.

“Based on the assumption that the experience of craving is independent of substance type, the team modified the PACS for drug craving measurement”.

2. The paper seems to contract its own definition of craving (paragraph at top of pg. 3).

Answer: Thank you for the suggestion. In the article we did not attempt to create our own definition. The whole paragraph (lines 46-54) was based on a literature review of the craving phenomenon. The issue has been clarified.

3. The introduction reads as somewhat meandering by discussing a host of topics related to craving. It would benefit from being streamlined to the topic at hand.

Answer: Thank you for the suggestion. We have carefully re-read the introduction and changed the structure of this part of the paper. The entire fragment (lines 55-63) about the correlation between craving and other variables has been removed from the Introduction. This description is a theoretical explanation for choosing variables constituting comparative criteria (see answer #7). Therefore we have used it at the beginning of the Criterion Validity section. Moreover, the fragment about the PACS characteristics (lines 77-85) has been moved to a new subsection titled Instruments.

In our opinion, all the changes made have streamlined the Introduction.

4.

A. The description of when and how the participants were ascertained is confusing (pg. 5, paragraph beginning “The research was conducted from…”). What is meant by starting “therapy”? Also, the manuscript says data was collected longitudinal, but the following sentence implies that different patients were assessed at different points in therapy.

Answer: Thank you for drawing attention to this issue. Two paragraphs (lines 101-114) have been clarified:

“The research was carried out in 14 inpatient and 13 outpatient randomly selected facilities for the psychosocial treatment of people with SUD in Poland. In most facilities participating in the research, the treatment was based on cognitive-behavioural psychotherapy, modified mostly by combining it with methods such as motivational interviewing, therapeutic community or solution-focused brief therapy. Average therapy duration was 6 months, with a range from 2 to 12 months.

The research was conducted from June 2018 until July 2019. Once records with missing data regarding responses to the PDCS questions were removed, the analyses were conducted on 282 cases. The data collection process was developed during two studies. In Study 1, data were collected from 111 patients at different stages of the therapy. Study 2 was longitudinal and consisted of measurements at two time points (T1 and T2). T1 was conducted among patients at the beginning of the therapy (where the beginning of therapy means that the patients had been under treatment in a particular facility for no longer than two weeks). 171 patients were surveyed at T1. At T2, data were collected from 70 out of these patients who had completed their therapy (the rest failed to complete the therapy)”. 

B. Was this psychosocial treatment only?

Answer: Yes, it was psychosocial treatment only, based on cognitive-behavioural psychotherapy modified mostly by combining it with methods such as motivational interviewing, therapeutic community or solution-focused brief therapy. This information had already been included in lines 103-105. Moreover, we have added the word “psychosocial” before “treatment”.

5.

A. In the “primary drug used” of Table 1, please provide what is being abbreviated by “NPS”.

Answer: Thanks for the suggestion. In Table 1 the full form of the abbreviation “NPS” has been added (New Psychoactive Substances).

B. Does primary drug use correspond to the substance use disorder (SUD) diagnoses? Would it be possible to provide SUD diagnosis as well as drug of choice?

Answer: The wording “Primary drug used” in Table 1 could be misleading. We have replaced the row header with “The most frequently used drug”. These data only concern the self-reported most frequently used substances. It is highly probable that they correspond to SUD diagnoses; however, we did not conduct analyses of diagnostic documents. So it is not possible to provide an SUD diagnosis as well as drug of choice. 

C. Mean & SD of length of treatment would also be appropriate to include in this table.

Answer: Thank you for your suggestion. We have added this information in Table 1.

6.

A. Given the length of treatment was highly variable (between 2-12 months), would this impact the LMI results being compared for T1 & T2? 

Answer: No, the variable length of treatment would not impact the LMI results. Despite the diversity of the data, the measures of model fit were very good. In addition, data were always collected for the LMI analyses – regardless of the length of therapy – at the beginning and at the end of therapy, so each patient completed the entire treatment program. Conclusions regarding this issue were included in the Discussion section.

B. Were the mid-therapy assessments not used in any analyses?

Answer: The description of the study procedure may have led to the misconception that data from the mid-therapy assessment (study 1) were collected longitudinally, which in fact was not the case. Only the data in study 2 – measurements T1 and T2 – were collected longitudinally (see answer #4), and these were used in the LMI analyses. On the other hand, data from patients during therapy (study 1) were used for: CFA, estimation of reliability coefficients, criterion validity and normalisation.

We hope that the changes in the Research procedures section (lines 101-114) have clarified this issue for the reader.

7.

A. The measures used to assess criterion validity are not discussed in the methods section.

Answer: Thank you for drawing attention to this omission. Guided by the reviewer’s advice, we have added a separate Instruments subsection. It contains information about the instruments used to assess criterion validity.

B. Besides the SPN, which seems to be another measure of craving, how were the other measures picked? How do they demonstrate criterion validity? There is no rationale provided as to why those measures were chosen. I wouldn’t necessarily expect there to be strong correlation between PDCS and the majority of the measures presented in Table 7.

Answer: The measures used to assess criterion validity were picked on the basis of the relapse prevention model. In this model, different variables, including craving, determine a risk of relapse. Therefore their mutual interactions are also assumed. This allowed us to treat the selected variables as the comparative criteria. Based on this theoretical concept, we did not expect strong correlations either. We assumed that all these variables – jointly determining the risk of relapse – should correlate with each other statistically significantly, although weakly or moderately. In response to this comment, we have modified the Criterion validity section, making use of a fragment from the Introduction (see answer #3), in the following way:

“The assessment of criterion validity is always based on an analysis of relations. In publications addressing the issue of a correlation between craving and predicting a relapse, craving is shown as a co-determining factor, alongside other intra- and interpersonal variables such as self-efficacy, motivation, negative affect (aggression, self-aggression – self-injury, impulsiveness) and social relations (sense of loneliness, social support). Most of the reported relations are incorporated in the cognitive-behavioural model of relapse. All of the listed factors contribute to a relapse; therefore their mutual interactions are also assumed. Based on this assumption, these variables were considered comparative criteria. The criterion validity assessment involved the measurement of the relations of the PDCS with instruments testing criterion variables and other scale assessing craving”.

In response to the comment on the strength of correlation, we have amended the passage in the text (lines 288-291).

8. In the discussion, further commentary on the strengths/limitations of the study is warranted.

Answer: Thank you for the suggestion. We have modified the Discussion section, highlighting the issue of strengths and limitations of the study.

9. The findings of the study are somewhat overstated in the conclusion section. While the PDCS may be used clinically, this study does not “confirm its clinical utility” nor investigate whether this measure improves treatment planning or predicts relapse.

Answer: Thank you for focusing attention on this overstatement. We have deleted this problematic fragment.

REVIEWER #2:

1. The Polish Drug Craving Scale (PDCS) should be mentioned in the title instead of PACS.

Answer: Thank you for the suggestion, but we have decided not to change the title of the manuscript. Our decision is based on previous arrangements with the Research Society on Alcoholism (RSA), which has exclusive copyrights to the PACS. Such wording of the title best protects the copyrights of the authors of the PACS and has been accepted by the RSA. The title expresses that the paper is about assessing the psychometric properties of the Polish version of the "PACS", modified for drug craving measurement. 

2. On page 4, some specific research questions can be developed to guide the reader to better understand the study. These questions can follow “Aims of the analysis”. In addition, “Aims of the analysis” can be renamed “Aims of This Study”.

Answer: Thank you for the suggestion. We have changed the title of the section Aims of the analysis to Aims of the study. With regard to specific research questions, we have decided not to add them. In our opinion, developing the research questions – where two hypotheses are included – could not improve understanding of the study.

3. In the methods, the authors missed the description of the PDCS (Table 3). How many items? How many points? Likert scale?

Answer: Thank you for pointing out this deficiency. In the Methods section, we have added an Instruments subsection. A detailed description of the PACS and the PDCS is provided in this subsection.

4. Line 148 on page 7, MPLUS needs a citation.

Answer: Thank you for drawing attention to this omission. It has been corrected.

5. Lines 155-156 on page 7, please specifically indicate what tests need the p-values?

Answer: Thank you for your comment. We have changed the sentence in lines 155-156 in relation to this issue: 

“For all analyses involving a probability value, 0.05 was assumed as the threshold for statistical significance. In the presentation of the results of analyses in which a p-value was needed, it was reported each time”.

Such a change seems sufficient to us.

6. Lines 157-158 on page 8, please specifically indicate what analyses using MPLUS and LAVAAN, respectively.

Answer: Thank you for this comment. In order to convey this specific information we have changed the sentence from lines 157-158 in the following way:

“The modelling was performed with Mplus 8.3. The reliability and criterion validity analysis were conducted using RStudio 1.2.5. with the application of the lavaan package. Furthermore, Jasp 0.12.2 statistical software was used for other analyses”. 

7. Some statistical analyses in the results were not mentioned in the methods. The authors reported criterion validity and percentile norms of the scale. But I could not find any statements regarding these two results in the methods.

Answer: Indeed, originally in the Data analysis subsection, there was no information about the normalisation method. We have corrected this issue by adding the following statements:

“A normalisation of the PDCS results – due to the skewed character of their distribution – was prepared using a tercile scale. A tercile scale does not reflect the shape of the raw score distribution; the distribution of its values is always uniform. This means there is the same probability of the occurrence of all values of a variable”.

Regarding criterion validity, in the initial version of the manuscript (subsection Data analysis – lines 152-154), the following information had already been included: 

“A criterion validity analysis was also conducted by determining the value of the r-Pearson correlation coefficient between the PDCS result and the results from other tools, constituting the comparative criteria”.

8. For criterion validity, I am not sure if the authors used latent scores or observed scores to correlate with criterion variables.

Answer: For criterion validity, observed scores were used. The r-Pearson correlation coefficient was calculated between the observed general score of the PDCS and the observed scores of the other scales constituting the comparative criteria. In order to address the issue more specifically in the manuscript, we have changed the sentence from lines 265-267 in the following way:

“This is indicated by statistically significant correlations between the observed general score of the scale and the observed results from other tools used”.

9. On page 8, since all the values are reported in the text, there is no need to present this table (Table 2). Please add 95% RMSEA in the text as well.

Answer: Thank you for the suggestion. Table 2 has been deleted and values of the 95% CI for RMSEA have been added in the text.

10. The longitudinal measurement invariance should be reported before the descriptive statistics at T1 and T2.

11. LMI results should be reported before the descriptive statistics. Because as the authors mentioned that “it is reasonable to compare latent variable means (drug craving) obtained during consecutive measurements” due to LMI of the scale.

Answer: Thank you for the suggestion, but we have decided not to change the sequence of the text. The arrangement of content in the text reflects the sequence of undertaken research and analysis activities. Before deciding to examine LMI, we analysed descriptive statistics for data from T1 and T2. Consequently, it was the results obtained, supplemented by the Wilcoxon signed-rank test, that provoked the question of whether the observed differences in craving levels were the effect of therapy or due to the lack of reliability over time of the PDCS. LMI was chosen as the statistical method to answer this question. Moreover, it seems to us that this sequence of content presentation allows readers – particularly readers not familiar with this method – to better understand the importance of the LMI in the context of assessing the reliability over time of a tool.

12. Lines 239-240, the statistical values for chi-square with df and p-value can be reported here.

Answer: Thank you for the suggestion. The mentioned statistical values have been reported.

13. Some statistics reports violate the APA style (p < .05 or p < .01 or p < .001).

Answer: Thank you for the comment. We checked our way of reporting statistics carefully. Consequently, we have changed the style of presentation of 95% CI (lines 32-33; 163-164). However, we have decided not to make changes in p-value reporting. In our opinion, the method of reporting these results used in the manuscript complies with the community standards contained in the PLOS ONE Submission Guidelines. Also, we have verified that the used style of presentation of the p-value (i.e. p<0.05 or p<0.01 or p<0.001) prevails in papers published in PLOS ONE.

---

## [Editor Report · Decision Letter 1]

29 Jul 2021

Psychometric properties and longitudinal measurement invariance of the drug craving scale: modification of the Polish version of the Penn Alcohol Craving Scale (PACS)

PONE-D-20-36075R1

Dear Dr. Opozda-Suder,

We’re pleased to inform you that your manuscript has been judged scientifically suitable for publication and will be formally accepted for publication once it meets all outstanding technical requirements.

Kind regards,

Frantisek Sudzina

Academic Editor

PLOS ONE
---

## [Editor Report · Acceptance letter]

27 Aug 2021

PONE-D-20-36075R1 

Psychometric properties and longitudinal measurement invariance of the drug craving scale: modification of the Polish version of the Penn Alcohol Craving Scale (PACS) 

Dear Dr. Opozda-Suder:

I'm pleased to inform you that your manuscript has been deemed suitable for publication in PLOS ONE. Congratulations! Your manuscript is now with our production department. 

Kind regards, 

on behalf of

Dr. Frantisek Sudzina 

Academic Editor

PLOS ONE